# From cylinder to city: How the recondensation-induced nucleation shapes urban aerosol number

Jen-Ping Chen<sup>1,2,3</sup>, I-Chun Tsai<sup>3</sup>, Li-Wei Kuo<sup>1</sup>, Gong-Do Hwang<sup>4</sup>

<sup>1</sup>Department of Atmospheric Sciences, National Taiwan University, Taipei City, Taiwan

<sup>2</sup>International Degree Program in Climate Change and Sustainable Development, National Taiwan University, Taipei City, Taiwan

<sup>3</sup>Research Center for Environmental Changes, Academia Sinica, Taipei City, Taiwan

<sup>4</sup>NVIDIA Singapore





Correspondence to: Jen-Ping Chen (jpchen@ntu.edu.tw)

**Abstract.** Air-quality models frequently underestimate fine particle number concentration (PNC), particularly in the nucleation/Aitken range—while reproducing PM<sub>2.5</sub> mass more accurately, suggesting that key number-forming processes are missing from current frameworks. We propose and investigate a physically motivated pathway, Recondensation-Induced Nucleation (RIN), in which pre-existing ambient aerosols are vaporized during combustion and subsequently re-nucleate as the exhaust cools, selectively boosting particle number with negligible impact on mass.

Controlled four-stroke engine experiments demonstrate that a distinct nucleation mode (<30 nm) appears only when ambient aerosols are present in the intake air, providing direct laboratory evidence of RIN. Parcel-model simulations of H<sub>2</sub>SO<sub>4</sub>–H<sub>2</sub>O systems were then used to probe particle evaporation under in-cylinder condition and the observed self-limiting behavior of nucleation. A parameterized RIN module was then implemented in the Community Multiscale Air Quality (CMAQ) model and tested over Taiwan. Without RIN, CMAQ underpredicted PNC by 75% and overpredicted PM<sub>2.5</sub> by 21% at the Xitun urban site; incorporating RIN reduced the PNC bias to 14% with negligible change in PM<sub>2.5</sub>. The RIN mechanism thus transfers accumulation-mode mass to Aitken-mode number, not only improve the low-PNC bias but also the low Aitken- to accumulation mode number ratio bias found at the Xitun site. Therefore, the inclusion of the RIN mechanism provides a mechanistic basis for improving health-relevant urban aerosol simulations.

#### 25 1. Introduction

The impact of aerosols on human health is a major concern in air pollution research. A growing body of evidence shows that mass concentration is not the sole or most appropriate measure of potential health effects, implying that health studies must consider other characteristics, such as particle number, particle morphology, and detailed chemical speciation (Mauderly 1993, 1999; Glovsky 1997; Albritton and Greenbaum 1998). Epidemiological and toxicological research indicates that the particle number concentration (PNC), especially of ultrafine particles (UFPs, 







UFPs contribute less to mass than larger particles but can penetrate deep into the respiratory system and even enter the bloodstream, leading to inflammation as well as respiratory and cardiovascular diseases (Pope III and Dockery 2006). For the same amount of mass, a high number concentration also implies a large surface area, which leads to stronger toxicity and chemical reactivity with body molecules (Oberdörster et al. 2005; Balmes and Hansel 2024). Several recent studies have highlighted the importance of PNC as a critical metric for assessing air pollution's health impacts (e.g., Kelly and Fussell 2020; Schraufnagel 2020; Schwarz et al. 2023; Stafoggia et al. 2023). Air quality and atmospheric chemistry modeling remains a crucial tool for both scientific research and policy-making concerning atmospheric aerosols. Modern models can now simulate the aerosol particle size distribution (PSD), which is a crucial property for assessing aerosol impact on human health and climate. While PNC is derived from this distribution, recent studies have found that models often underestimate PNC (Xausa et al. 2018; Fanourgakis et al. 2019; Leinonen et al. 2022; Kohl et al. 2023; Wang et al. 2023; Kim et al. 2025), particularly in the nucleation and Aitken modes. Such underestimation arises from various factors, including but not limited to limitations in emission inventories (Wang et al. 2023), emission PSD (Park et al. 2005; Spracklen et al. 2010; Sartelet et al. 2022), missing/weak nucleation pathways (Kusaka et al. 1995; Yu 2001; Sorokin et al. 2005; Young et al. 2008; Dunne et al. 2016; Wang et al. 2023; Shao et al. 2024), simplifications in various microphysical processes (Chen et al. 2011; Mann et al. 2012; Westervelt et al. 2014; Chen et al. 2021; Patoulias and Pandis 2022), and numerical methods (Mann et al. 2012; Jacquot and Sartelet 2025). In air quality simulation over urban areas, where the health issue is most critical, the underestimation of PNC is unlikely to be caused by weak/missing nucleation pathways, because the high concentration of existing aerosols will efficiently deplete precursors (e.g., sulfuric acid), leaving the nucleation process inconsequential. Moreover, other possibilities mentioned above also lack direct verification for the underestimation of PNC in urban areas. Among these potential model deficiencies, this study investigated a new particle production mechanism proposed by Chen (1999), which is related to both emission and nucleation in urban areas. This Recondensation-Induced Nucleation (RIN) mechanism suggests that aerosols pre-existing in the intake air may be vaporized during engine combustion, and the vapors thus generated may produce strong nucleation during cooling of the exhaust air. Chen (1999) provided supporting

human health than mass concentration (Oberdürster 2000; Chung et al. 2015; Li et al. 2016b). Due to their diminutive size,

Organization of the paper:

implementing them in a regional air-quality model.

Section 2 describes the engine-exhaust experiments, including the setup and key results.

experimental evidence by heating up ambient air through a high-temperature furnace and then immediately cooling it down, revealing a strong production of UFP at the expense of existing accumulation-mode aerosol particles. However, the furnace experiment by Chen (1999) did not involve real combustion, which on its own may also generate UFP, meaning the results cannot be directly applied in air quality models. Therefore, one of the main objectives of this study is to extend the work of

Chen (1999) to more realistic conditions by conducting engine combustion experiments with and without ambient aerosols in the intake air. We further evaluate the impact of RIN on PNC in the urban area by parameterizing the laboratory results and

- Section 3 presents the aerosol parcel—model simulations used to interpret the experimental findings.
  - Section 4 parameterizes the experimental results and implements them in a regional air-quality model to assess the overall impact.
  - Section 5 summarizes and concludes with the main findings.

# 2. Engine Combustion Experiments

We conducted engine-exhaust experiments under three intake conditions—Amb-In, Pur-In, and Lad-In—and measured particle size distributions (11.5–453 nm) with a TSI SMPS 3934.

# 2.1 Experimental setup

The most common motor vehicles in the urban areas of East and Southeast Asian countries are gasoline-engine motorcycles (including scooters) and passenger cars. Because testing high-displacement passenger-car engines on our bench was impractical, we used a four-cylinder motorcycle engine for the combustion experiments. We hypothesize the RIN mechanism operates similarly across vehicle types as their peak burned-gas temperature is broadly comparable and sufficient to volatilize pre-existing aerosol particles. Accordingly, we apply the experimental results to both motorcycles and passenger cars in the model simulations.

The combustion experiments were conducted with intake air under controlled aerosol conditions, according to the schematic shown in Fig. 1. We consider three types of experiments: (1) Intake of ambient air (Amb-In), (2) Intake of purified air (Pur-In), and (3) Intake of artificial aerosol-laden air (Lad-In). The Amb-In versus Pur-In contrast isolates and demonstrates the RIN mechanism under realistic conditions, extending the conceptual experiment of Chen (1999). As Chen (1999) suggested that the intensity of new particle formation via RIN depends on pre-existing particle loading, the Lad-In experiments were designed to explicitly test this dependence by varying the seed aerosol mass/number.

85

 $Figure \ 1. \ Engine-exhaust \ experiment \ flow \ paths \ (Amb-In \ / \ Pur-In \ / \ Lad-In) \ with \ SMPS \ sampling; \ blue \ arrows \ indicate \ flow; \ valves \ and \ buffer \ bags \ annotated.$ 

- To provide intake air for experiment Pur-In, we applied laboratory-grade purified air and verified its particle-free status prior to use. We also considered using a high efficiency-particulate air (HEPA) filter for removing particles, but rejected this option because condensable gases (e.g., ammonia and nitric acid) can pass through and influence nucleation in the exhaust air. To mimic ambient moisture, the purified air was humidified to ~85% relative humidity (*RH*) by passing it through a 4.0 mol kg<sup>-1</sup> salt solution before entering the engine.
- For the Lad-In experiments, the intake air was first filtered with a HEPA filter and then seeded with aerosol generated from ammonium sulfate solution (in deionized water). Unlike the Pur-In case, residual condensable gases upstream are not critical here because the loaded aerosol evaporates in-cylinder, producing far greater amounts of condensable vapors that drive nucleation upon cooling. The seed solution had a molality of 4.455 mol kg<sup>-1</sup>, chosen to be roughly in equilibrium with ~85% RH. Particles were produced with a TSI 9302 Single Jet Atomizer, yielding a modal diameter of ~100 nm (cf. Fig. S1).
  - Ammonium sulfate was selected to represent local aerosol composition (prevalent in the study region) and because its neutralized, less corrosive nature (relative to sulfuric acid) enables safer handling. The PNC introduced into the engine was controlled by diluting the aerosol with purified air. All intake-air streams (Amb-In, Pur-In, and Lad-In) were temporarily held in a 200 L buffer bag for flow stabilization before entering the combustion engine. Buffer bags and tubing were replaced frequently to prevent cross-contamination, and each bag was purged with filtered air at the start of every experiment.
  - The combustion platform was a 100 cc, four-stroke motorcycle engine with electronic fuel injection (EFI) operated at a steady 2000 rpm. The engine was warmed for 20 min before each run to reach thermal and emissions steady state. Peak burned-gas temperatures in such engines typically reach ~1700–2600 K during ignition and combustion of the air–fuel mixture (Shehata 2010; Mosburger et al. 2013; Heywood, 2018), although temperature at the cylinder walls and piston surface is much lower, often controlled to about 450 K in car engines due to cooling systems. These temperatures far exceed




the nominal boiling point of sulfuric acid ( $\sim$ 444.7 K) and the decomposition/vaporization threshold of ammonium sulfate ( $\sim$ 510  $\pm$  20 K), enabling partial to near-complete in-cylinder evaporation of pre-existing particles. The subsequent exhaust cooling thus favors RIN.

The particle size distributions (PSDs) were measured with a TSI SMPS 3934 over 11.5–453 nm. In our earlier experiments, particle sampling was conducted directly in the open air behind the tailpipe; however, the measured concentrations exhibited considerable fluctuations due to pressure pulsations from the engine cylinder and turbulent mixing with ambient air. To reduce this variability, the exhaust was routed into a buffer bag and then drawn continuously into the SMPS. In this revised setup, the exhaust air primarily cools through conduction along the walls of the exhaust pipe, buffer bag, and sampling tubes, while mixing with ambient air was minimized and not considered in the analysis. Each experiment was repeated multiple times (n = 7 for Experiments 1 and 2;  $n \ge 4$  for each case in Experiment 3), and we report the run-averaged PSDs. Ambient PSDs were measured immediately before and after each experiment to document background conditions.

## 2.2 Experimental results

Figure 2: Comparison of PSD for Amb-In, Pur-In, and ambient intake-air treatments. Amb-In shows distinct <30 nm nucleation mode and reduced accumulation number.

The Amb-In vs. Pur-In comparison (Fig. 2) clearly demonstrates the RIN mechanism: PNC in Amb-In is substantially higher than in Pur-In, and the Amb-In PSD exhibits a prominent nucleation mode (< ~30 nm). In Pur-In, particles in the exhaust are expected to consist primarily of combustion residuals (e.g., soot and organic fragments). These particles are unlikely to be ambient particles introduced through mixing or leakage, as the buffer-bag design prevents entrainment of outside air into the system. A small fraction of fuel sulfur (~10 ppm by mass) may oxidize to H<sub>2</sub>SO<sub>4</sub> (e.g., Sorokin et al. 2004; Fleig et al. 2013), but the feeble nucleation mode observed in Pur-In indicates that fuel-sulfur-driven nucleation is minor under our conditions. In contrast, Amb-In introduces pre-existing ambient aerosol that evaporates in-cylinder and re-nucleates as the exhaust cools,





yielding the strong nucleation-mode peak and, notably, a marked reduction of accumulation-mode number relative to ambient air. This pattern is consistent with mass-to-number conversion via RIN—i.e., condensation of evaporated material onto many newly formed small particles rather than onto fewer, larger ones.

We also note that the accumulation-mode number in Amb-In is lower than in Pur-In, even though both conditions should generate soot. This could reflect suppressed soot formation (e.g., altered flame chemistry or cooling rates in the presence of evaporating ambient aerosols) or subtle procedural differences (e.g., flow paths indicated in Fig. 1). Disentangling these possibilities requires targeted diagnostics and is left for future work.

The Lad-In experiments show that exhaust PNC increases with intake aerosol loading (Fig. 3), corroborating Chen (1999) that RIN-driven particle production depends on the abundance of pre-existing aerosols. However, the output PNC varies by only about one order of magnitude despite a three-order-of-magnitude change in input loading. This muted response likely reflects (i) a baseline soot contribution present across all loadings that dilutes the apparent RIN sensitivity, (ii) self-limiting nucleation: as numerous new particles form, enhanced condensation and coagulation sinks suppress further number growth (Lehtinen et al. 2007; Tuovinen et al. 2022), and (iii) the limitation in certain nucleation pathways, chemiion–mediated nucleation (Yu et al 2004). In addition, at very high loadings, incomplete in-cylinder evaporation of the seeded aerosol may cap the vapor available for re-nucleation, further weakening the scaling. We examine these two factors with parcel-model simulations in the next section.

Notably, Lad-In exhaust PNCs exceed those from Amb-In. Beyond differences in experimental plumbing and ambient conditions (e.g., tubing residence time affecting growth before SMPS sampling), this may indicate that a substantial fraction of ambient aerosol is a poor nucleation precursor (e.g., organic carbon), as suggested by regional composition measurements (Chou et al. 2010, 2022).

Figure 3: Exhaust PNC increases with intake aerosol loading.

## 3. Idealized Air Parcel Simulation

To interpret the laboratory results, we developed an aerosol parcel model (H<sub>2</sub>SO<sub>4</sub>–H<sub>2</sub>O nucleation, condensation, coagulation) that resolves heating and cooling histories.

# 3.1 Model setup







The parcel model used here is adapted from Chen et al. (2011), which employed a multi-component particle framework classifying aerosols by their water and sulfate mass. Guided by observations in our study region—where Cheung et al. (2013, 2016) found that new particle formation is dominated by H<sub>2</sub>SO<sub>4</sub>–H<sub>2</sub>O nucleation—we idealize the system to sulfuric acid and water only. The model includes binary nucleation (H<sub>2</sub>SO<sub>4</sub>–H<sub>2</sub>O), vapor condensation of both species (with solute/Raoult and curvature/Kelvin effects), and Brownian coagulation. The nucleation scheme follows the classical binary nucleation theory, with an additional correction for the curvature dependence of surface tension (Chen et al., 2011). Conventional binary nucleation schemes tend to underestimate nucleation rates, whereas incorporating the curvature effect substantially improves performance, producing PNC values much closer to observations, as demonstrated by Chen et al. (2011). Each chemical component is discretized into 45 mass bins. For water, the bin range spans  $1.0 \times 10^{-24}$  to  $1.0 \times 10^{-9}$  mol; for sulfate,  $1.0 \times 10^{-24}$  to  $5.6 \times 10^{-11}$  mol. Changes in the particle spectrum along the mass coordinates—arising from condensation/evaporation and Brownian coagulation—are solved using a moment-conserving advection scheme that preserves both total mass and particle number across the discretized bins. We simulated both in-cylinder heating and post-combustion cooling of an air parcel with idealized settings. For the heating phase, the parcel initially contains sulfate aerosol with geometric standard deviation ( $\sigma_a$ ) and geometric mean diameter ( $D_a$ ) prescribed from the urban aerosol PSD of Whitby (1978). Following typical engine transients (Heywood 2018), the temperature is ramped from 298 K to 800 K over 5 ms, representing rapid heating to the exhaust-header regime. It is then held at 800 K to assess particle survival under hot exhaust conditions. A temperature of 800 K corresponds approximately to the end-of-compression condition and therefore represents a lower bound of heating. In practice, simulations at temperatures above 800 K are largely unnecessary, however, because most particles are expected to fully evaporate at such elevated temperatures. The heating simulation uses a 0.1 ms time step throughout.

The cooling stage begins at T = 383 K (110 °C), below which the nucleation rate is negligible. Exhaust cooling is assumed to occur purely by conduction through the tailpipe, and we parameterize it with Newton's law of cooling using two measurements: an exhaust temperature of  $\sim 333$  K at 15 cm downstream the tailpipe and a  $\sim 2$  s travel time from cylinder to tailpipe exit. These constraints set the exponential cooling rate applied in the parcel model. The cooling simulation uses a 0.01 s time step. We further assume that a specified fraction of the seeded aerosol survives heating (i.e., does not fully evaporate). The residual particle sizes should decrease as they evaporate. However, the size spectrum of the residual particles depends not only on the initial loading but also on the duration and rate of evaporation, which are highly uncertain. To avoid introducing poorly constrained assumptions, we adopted a simplified approach in which the surviving particles retain the





same chemical composition and size distribution parameters ( $\sigma_g$  and  $D_g$ ) as initially prescribed, but with a reduced number concentration. It should be noted that the simulations do not account for wall losses of vapors or particles, which are likely to occur in the laboratory experiments.

## 3.2 Simulation results

Figure 4a shows that, by the end of the combustion heating the particle number concentration drops by nearly four orders of magnitude. Residual particles may remain, but they are negligible unless the initial aerosol loading is very high. At 800 K, most survivors persist for only a few milliseconds. The evaporation rate is regulated by aerosol loading: higher loadings release more H<sub>2</sub>SO<sub>4</sub> (and/or SO<sub>3</sub>) vapor, elevating its partial pressure and thereby slowing further evaporation. Consequently, more residual particles are expected at higher initial loadings, as seen in Fig. 4a. We note, however, that 800 K exceeds water's critical temperature (647 K) and approaches the estimated critical temperature of sulfuric acid (~927 K), so the thermodynamic constraints are uncertain. To bracket this uncertainty, subsequent parcel simulations prescribe several evaporation fractions and assess the sensitivity of the residual-particle effect.

Figure 4: Simulated time evolution of particle concentration. (a) Heating: rapid evaporation; (b) Cooling: burst nucleation with sensitivity to the assumed evaporation fraction. Blue, green, and red curves correspond to initial aerosol number concentrations C= 10<sup>3</sup>, 10<sup>4</sup>, and 10<sup>5</sup> cm<sup>-3</sup>, respectively. Solid and dotted lines denote 100% (Evp1.0) and 10% (Evp0.1) evaporation during the heating phase. The black dash—dot curve shows the parcel temperature (right-hand axis). The small zigzag pattern in panel (a) arises from numerical discretization of mass bins and is not a physical signal.

Figure 4b illustrates the time evolution of particle number concentration (PNC) during the cooling stage (for clarity, only the 100% and 10% evaporation cases are shown). As the parcel cools sufficiently, new particle formation is observed as a nucleation burst. This burst is directly triggered by the rapid increase in relative humidity (RH) and relative acidity (RA), defined as the ratio of H<sub>2</sub>SO<sub>4</sub> vapor pressure to its saturation value, as detailed in Fig. 5. The burst initiated sooner when either the preexisting aerosol loading is higher or the evaporation fraction is larger, demonstrating its sensitivity to the availability of the H<sub>2</sub>SO<sub>4</sub> vapor and RA. Following the onset, PNC rapidly rises to a peak and subsequently declines, a trend




indicative of a self-limiting process: newly formed particles consume precursor vapors by condensation (which correlates with the rapid decrease of RA shown in Fig. 5), curbing further nucleation, and thereafter coagulation reduces particle number. For low evaporation fractions (e.g., 10%), the peak PNC increases with the input aerosol loading. Under 100% evaporation, however, the peak varies only weakly with loading; in fact, the peak is slightly higher for C= 10<sup>4</sup> cm<sup>-3</sup> than for either 10<sup>3</sup> cm<sup>-3</sup> or 10<sup>5</sup> cm<sup>-3</sup>. This non-monotonic response further underscores the self-limiting nature of nucleation—especially at high initial loading and high evaporation fraction—where enhanced condensation and coagulation sinks dampen the growth of particle number despite abundant vapor.

Figure 5: Time evolution of relative humidity (defined with respect to  $H_2O$ ) and relative acidity (defined with respect to  $H_2SO_4$  vapor) during the cooling stage (corresponding to the PNC evolution in Fig. 4b). Relative acidity (RA) is categorized by evaporation fraction (thick red curves: 100%; thin red curves: 10%) and by initial PNC (dashed, dotted, and solid curves correspond to  $C = 10^3$ ,  $10^4$ , and  $10^5$  cm<sup>-3</sup>, respectively). The initial mole fraction of  $H_2SO_4$  vapor corresponding to  $C = 10^3$ ,  $10^4$ , and  $10^5$  cm<sup>-3</sup> are  $2.5 \times 10^{-9}$ ,  $2.5 \times 10^{-8}$ , and  $2.5 \times 10^{-7}$ , respectively, assuming 100%. evaporation.

Figure 6 summarizes how the peak PNC varies with initial aerosol loading and evaporation fraction. Peak PNC increases sharply when both parameters are small, but the sensitivity progressively saturates at higher loadings and higher evaporation fractions, consistent with strengthening condensation/coagulation sinks. This saturation explains the relatively weak dependence of exhaust PNC on intake loading seen in Fig. 3, especially under high evaporation conditions.

Figure 6: Peak PNC (applicate) produced by RIN from parcel simulations under varying input aerosol concentration (ordinate) and evaporation fraction (abscissa).

# 4. Regional Air Quality Simulations

We implemented a RIN parameterization in CMAQ and evaluated impacts on Aitken-mode PNC and PM<sub>2.5</sub> over Taiwan, with observations at Xitun (Taichung).

# 4.1 Model setup




The Community Multiscale Air Quality (CMAQ) model, v4.7.1 (Byun and Schere 2006) was applied here for regional simulations. The gas—aerosol chemistry was augmented to include advanced aerosol processes—notably secondary organic aerosol formation and particle coagulation (Tsai et al. 2015a,b; Li et al. 2016a). Meteorological fields driving CMAQ were produced with the Weather Research and Forecasting (WRF) model, v3.7.1 (Skamarock et al. 2008). The modeling domains span East Asia (15°–36° N, 100°–140° E) to capture large-scale influences (East Asian monsoon, tropical cyclones) and major source regions. We employed two-way nesting with an outer domain at 10 km horizontal resolution and an inner Taiwan domain at 2 km (Fig. 7). The vertical grid comprises 37 layers with enhanced resolution near the surface to better resolve turbulent mixing and vertical transport. Additional CMAQ and WRF configuration details are summarized in Table 1.

Figure 7: Map of CMAQ simulation domains. Xitun site marked.

Table 1: Model setup options for the CMAQ and WRF models.

| Model | Modules                                                                                                      |  |  |  |  |
|-------|--------------------------------------------------------------------------------------------------------------|--|--|--|--|
| CMAQ  | Gas-phase chemistry mechanisms: SAPRC99 (Carter 2000)                                                        |  |  |  |  |
|       | New particle formation mechanism: Kulmala et al. (1998)                                                      |  |  |  |  |
|       | Anthropogenic emissions                                                                                      |  |  |  |  |
|       | East Asia Emission Inventory: Multi-resolution Emission Inventory for China (MEIC), gridded at a 0.1°        |  |  |  |  |
|       | × 0.1° spatial resolution (http://meicmodel.org.cn)                                                          |  |  |  |  |
|       | Taiwan Emission Inventory: Taiwan Emission Data System (TEDS) version 10, gridded at a 1 km × 1              |  |  |  |  |
|       | km spatial resolution (https://air.moenv.gov.tw/airepaEn/EnvTopics/AirQuality_4.aspx)                        |  |  |  |  |
|       | Biogenic emissions:                                                                                          |  |  |  |  |
|       | Model of Emissions of Gases and Aerosols from Nature (MEGAN; Guenther et al. 2006), driven                   |  |  |  |  |
|       | by MODIS land cover and leaf area index (LAI) datasets.                                                      |  |  |  |  |
| WRF   | Initial and boundary conditions:                                                                             |  |  |  |  |
|       | ERA5 reanalysis data with a $0.25^{\circ} \times 0.25^{\circ}$ horizontal resolution (Hersbach et al. 2016). |  |  |  |  |
|       | Physics options:                                                                                             |  |  |  |  |
|       | Goddard cloud microphysics scheme (Tao et al. 1989)                                                          |  |  |  |  |
|       | Yonsei University planetary boundary layer scheme (Hong et al. 2006)                                         |  |  |  |  |




RRTMG shortwave and longwave radiation schemes (Iacono et al. 2008)

The unified Noah land surface model (Tewari et al. 2004)

The Kain-Fritsch cumulus parameterization (Kain 2004), applied only to the outer domain

# 4.2 RIN mechanism setup

The RIN mechanism effectively converts ambient sulfate mass into new Aitken-mode particles. Based on the laboratory results presented in Section 3, the ambient sulfate mass contained in the intake air is assumed to evaporate completely under in-cylinder conditions. We estimate the conversion rate by the volume of ambient air processed by on-road engines—i.e., intake airflow multiplied by the ambient sulfate mass concentration. Intake airflow is linked to the fuel consumption rate via the air–fuel ratio (AFR). We assume a stoichiometric AFR = 14.7:1 (air:fuel by mass), typical of spark-ignition vehicles. The fuel consumption rate is inferred from the carbon emission rates (CO + CO<sub>2</sub>) in the emission database using the stoichiometry of gasoline combustion (approximated as C<sub>8</sub>H<sub>18</sub>), following Heywood (2018). This approach yields a consistent, inventory-driven estimate of the ambient air volume processed—and thus the mass available for RIN-driven conversion into Aitken-mode number.

In addition to conserving the total mass of ambient aerosols, it is also necessary to define the size distribution of newly formed particles to accurately simulate the effects of the RIN mechanism. In CMAQ, aerosol physical properties are represented using a tri-modal particle size distribution (PSD) that includes the Aitken, accumulation, and coarse modes. The nucleation mode discussed earlier is not explicitly distinguished from the Aitken mode in CMAQ. Each of the three modes is characterized by a log-normal distribution expressed as:

$$n(x) = \frac{N}{\sqrt{2\pi}a} \exp\left[-\frac{1}{2}(\frac{x-\mu}{a})^2\right]$$
 (1)

where x = lnD is the natural logarithm of particle diameter (D), N is the particle number concentration,  $\mu = lnD_g$  is the modal value and  $\sigma = ln\sigma_g$  is the standard deviation. The parameters  $D_g$  and  $\sigma_g$  are referred to as the geometric mean diameter and geometric standard deviation, respectively.

In CMAQ, the Aitken mode aerosol emission is assumed to have a  $D_g$  of 13 nm and  $\sigma_g$  of 1.7 for fine particles containing sulfate, nitrate, and elemental carbon (Elleman and Covert, 2010). The nucleation mode shown in Figure 2 from the Amb-In experiment can be can be approximated by a lognormal distribution with  $D_g=18$  nm and  $\sigma_g=1.16$ . However, the particles observed in our measurements were likely hygroscopically grown, leading to significantly larger wet diameters than the dry particle sizes required for the CMAQ emission parameterization. In fact, the relative humidity of the exhaust air within the buffer bag (cf. Fig. 1) was likely elevated, as condensation on the bag walls was frequently observed. Previous studies have demonstrated that the wet particle diameter can exceed the dry diameter by several times under relative humidity conditions greater than 90% (Chen 1994; Achtert et al. 2009; Zieger et al. 2017; Chen et al. 2022). Here, we estimate the swelling ratio—defined as  $r_{eq}/r_d$ —using the diagnostic formula proposed by Chen et al. (2013) as

follows:






$$r_{eq}/r_d = [1 + \kappa/(a_1 + a_2/RH + a_3/r_d)]^{1/3}$$
(2)

where  $r_{eq}$  denotes the equilibrium (wet) radius,  $r_d$  the dry radius,  $\kappa$  the hygroscopicity, and  $a_1 = -1.02733$ ,  $a_2 = 1.02654$ , and  $a_3 = 6.07891 \times 10^{-10}$  m are empirical fitting coefficients. The newly formed particles are likely composed primarily of water and sulfuric acid, which facilitate efficient binary nucleation. It is also possible that trace gases such as ammonia, volatilized from ambient aerosols, re-condense upon new particle formation. Accordingly, we estimate the swelling ratio by assuming the particles consist of either sulfuric acid or ammonium sulfate, adopting  $\kappa=1.19$  for the former and  $\kappa=0.53$  for the latter (Petters and Kreidenweis 2007). For a possible RH range of 90% to 100%, the swelling ratio  $r_{eq}/r_d$  derived from equation (2) for sulfuric acid particles ranges from 1.97 to 2.48, corresponding to  $D_g$  of 9.15 to 7.25 nm. For ammonium sulfate particles, the derived ratios are  $r_{eq}/r_d = 1.61$  to 2.06, yield  $D_g = 11.2$  to 8.75 nm.

For our simulations, we selected  $D_g = 13$  nm (the default CMAQ configuration) and  $D_g = 7.25$  nm from the lower-bound of value range estimated above. The latter value represents conditions under high relative humidity. For  $\sigma_g$ , we adopted  $\sigma_g = 1.7$  (the default CMAQ setup) and  $\sigma_g = 1.2$  based on our experimental observation (rounded to the first digit). It should be note that  $\sigma_g$  of a dry spectrum may differ slightly from the that of the wet spectrum shown in Fig. 2, because the swelling ratio varies with particle sizes according to equation (2). However, accurately determining  $\sigma_g$  of the dry spectrum is challenging.

A control simulation (referred to as NoRIN) was performed with the RIN mechanism disabled to establish baseline comparisons. The combinations of  $D_g$  and  $\sigma_g$  values were used to design sensitivity experiments for evaluating the possible impact of the RIN mechanism on aerosol number concentrations (Table 2). The RIN<sub>CTRL</sub> case represents the default CMAQ aerosol emission configuration, while RIN<sub>D</sub>, RIN<sub> $\sigma$ </sub>, and RIN<sub>D $\sigma$ </sub> denotes simulations with variations in the parameters  $D_g$  (denoted by subscript D) and  $\sigma_g$  (denoted by subscript  $\sigma$ ). The NoRIN experiment thus serve as a reference case without the RIN mechanism.

Table 2: Comparison of mean PM<sub>2.5</sub> and PNC across experiments and observation. The values are average over the whole simulation period in November 2020.

|                     | Geometric<br>Standard<br>Deviation | Modal<br>Diameter<br>(nm) | Average<br>PM <sub>2.5</sub><br>(μg m <sup>-3</sup> ) | Average<br>PNC<br>(cm <sup>-3</sup> ) |
|---------------------|------------------------------------|---------------------------|-------------------------------------------------------|---------------------------------------|
| Observation         |                                    |                           | 16.75                                                 | 9306                                  |
| NoRIN               |                                    |                           | 20.31                                                 | 2304                                  |
| RIN <sub>CTRL</sub> | 1.7                                | 13                        | 20.66                                                 | 2709                                  |
| $RIN_D$             | 1.7                                | 7.25                      | 20.29                                                 | 4163                                  |
| $RIN_{\sigma}$      | 1.2                                | 13                        | 20.26                                                 | 3366                                  |
| $RIN_{D\sigma}$     | 1.2                                | 7.25                      | 20.41                                                 | 7249                                  |

## 4.3 Aerosol measurement data

For verification of particle number concentrations (PNC) in the simulation results, we utilized hourly particle size distribution (PSD) data measured during a non-operational mission at the Xitun air quality monitoring station (24°09′41.4″ N, 120°37′02.6″ E), conducted by the Ministry of Environment. The Xitun station is located near the center of Taichung City, the second-largest metropolitan area in Taiwan. The instruments used for PNC measurement included a GRIMM Scanning Mobility Particle Sizer (SMPS+C; GRIMM Aerosol Technik Ainring GmbH & Co. KG, Germany), which measures particle diameters ranging from 5 to 350 nm across 117 size channels, and a GRIMM Environmental Dust Monitor (EDM 180), which covers the 0.25–32 μm size range with 31 channels. In addition, hourly PM<sub>2.5</sub> mass concentrations routinely obtained from a Beta Attenuation Mass Monitor (BAM 1020; Met One Instruments Inc.) at the Xitun station were also analyzed for comparison.

Considering the availability of PNC measurements, simulations were conducted for the period of November 5–25, 2020.

During this period, a total of 54 hours of PNC data were missing—primarily on November 15–16—and only 5 hours of PM<sub>2.5</sub> data were unavailable. These data gaps were excluded from the subsequent analyses.

## 4.4 Simulations results


On average, the NoRIN run overestimated PM<sub>2.5</sub> by 21% while underestimated PNC by 75% (Table 2). Introducing the RIN mechanism had little effect on PM<sub>2.5</sub> but substantially improved PNC, reducing the underestimation to 71% in RIN<sub>CTRL</sub>. The results were sensitive to the assumed PSD parameters. With  $D_g$ =7.25 nm (RIN<sub>D</sub>), the PNC bias reduced to 55%; with  $\sigma_g$  = 1.2 (RIN<sub> $\sigma$ </sub>), it decreased to 64%. Replacing both parameters simultaneously (RIN<sub>D $\sigma$ </sub>) further reduced the PNC








underestimation 14%.

The opposite-signed biases in PNC and PM<sub>2.5</sub> suggest the presence of a missing process that influences particle number and mass differently. This pattern rules out emission inventory or meteorological uncertainties—such as errors in advection or turbulent mixing—as the primary cause of the PNC shortfall, since these factors would typically affect both particle number and mass in the same direction. In contrast, the RIN mechanism intrinsically enhances particle number concentrations without a commensurate increase in mass, thereby reproducing the observed bias pattern.

Because the RIN mechanism is strongly associated with traffic activity, which exhibits a distinct diurnal variation, we analyzed the hourly evolution of aerosol properties on two representative days (November 20 and 25). Both days were characterized by fair weather conditions under mild northeasterly monsoon flow, providing stable regional background

conditions suitable for evaluating RIN-related variations. As shown in Figs. 8a and 8b, the simulated PM<sub>2.5</sub> concentrations were approximately twice the observed values during the early morning hours, likely due to a low bias in the nighttime planetary boundary layer height (PBLH). This bias is a well-documented issue in CMAQ simulations (e.g., Du et al. 2020; Cheng et al. 2021; Chuang et al. 2023) and is more pronounced on November 25. In contrast, PM<sub>2.5</sub> tends to be underestimated from late afternoon through evening. The morning PM<sub>2.5</sub> peak associated with rush-hour traffic is clearly reproduced, although the timing of the peak may be slightly shifted. However, the nighttime peaks observed in the measurements are not evident in the simulations for either day, likely reflecting deficiencies in simulating the timing and intensity of land-breeze development. In addition, the discrepancies may also arise from inaccuracies in the emission inventory, particularly regarding the diurnal emission profiles (Tsai et al. 2021; Chen et al. 2024). These modeling limitations, however, are beyond the scope of the present study and are not further discussed here.

The contrasting biases in PM<sub>2.5</sub> and PNC observed in the monthly averages (Table 2) become even more apparent on the selected analysis days. As shown in Figs. 8c and 8d, the NoRIN simulation produced substantially underestimated PNC values, despite exhibiting a positive bias in PM<sub>2.5</sub>. The simulated morning PNC peak was less than one-quarter of the observed value, and the discrepancy was even greater during the evening hours on both days. Incorporating the RIN mechanism led to notable improvements in PNC performance. For example, RIN<sub>CTRL</sub> increased the morning peak PNC by approximately one-third, while RIN<sub>σ</sub> doubled this improvement and RIN<sub>D</sub> tripled it, resulting in a peak magnitude approaching the observed value.

When both parameters— $D_g$  and  $\sigma_g$ —were replaced with those derived from our experimental measurements (RIN<sub>D $\sigma$ </sub>), the morning peak PNC appeared overestimated. However, considering that the simulated PM<sub>2.5</sub> peak was approximately twice the observed value, and that RIN-induced PNC production scales with ambient PM<sub>2.5</sub>, the RIN<sub>D $\sigma$ </sub> results remain physically consistent. Moreover, in the RIN<sub>D $\sigma$ </sub> case, the behaviors of PM<sub>2.5</sub> and PNC were mutually consistent—both were overestimated in the morning and slightly underestimated in the evening. In contrast, other RIN configurations continued to produce low PNC values despite the morning overestimation of PM<sub>2.5</sub>, underscoring the sensitivity of the RIN mechanism to particle size distribution parameters.

It is important to note that new particle production via the RIN mechanism is closely linked to traffic activity. Consequently,



even though PM<sub>2.5</sub> concentrations are elevated during the early morning hours—particularly on November 25—the PNC remains low until approximately 4–5 a.m., when traffic emissions begin to increase.

Notably, the modeled PNC peak precedes the PM<sub>2.5</sub> peak, a behavior consistent with observational findings reported by Chen (1999). Moreover, the rapid decline in simulated PNC also occurs earlier than that of PM<sub>2.5</sub>, which is likely attributable to self-limiting effects associated with the coagulation sink, as discussed previously.

Figure 8: Comparison of simulated and observed hourly variations of PM<sub>2.5</sub> (top) and PNC (bottom) on the two selected days. The simulated PM<sub>2.5</sub> values are from the RIN<sub>CTRL</sub> experiment, as results from other RIN cases are nearly identical. Observed values (OBS) are shown as black solid lines, while simulations from NoRIN, RIN<sub>CTRL</sub>, RIN<sub>D</sub>, RIN<sub> $\sigma$ </sub>, and RIN<sub>D $\sigma$ </sub> are represented by red dotted, blue solid, light green dotted, dark green dash-dotted, and cyan solid lines, respectively.

Figure 9 presents the simulated spatial distribution of PNC over Taiwan, averaged for November 20, 2020. A pronounced contrast is evident between the NoRIN and RIN<sub>D $\sigma$ </sub> scenarios, particularly in the partitioning between the Aitken and accumulation modes. In the absence of the RIN mechanism, aerosol particles are predominantly concentrated in the accumulation mode. In contrast, when the RIN mechanism is included, Aitken-mode particles dominate the total PNC. The PSD measurements at Xitun on the same day (see Supplement Fig. S2) support the RIN<sub>D $\sigma$ </sub> results, showing that the Aitken mode particles (

seven). The  $RIN_{D\sigma}$  simulation produced a comparable Aitken-to-accumulation ratio of about four to five, demonstrating reasonable agreement with the observations.

Figure 9 also illustrates that the RIN mechanism generally reduces accumulation-mode PNC, though the absolute magnitude of this decrease is modest. This limited reduction is likely due to replenishment through condensational growth and coagulation of newly formed particles. The spatial patterns for November 25, as well as those from other RIN simulations (RIN<sub>CTRL</sub>, RIN<sub>D</sub>, and RIN<sub>σ</sub>) are qualitatively similar and therefore not shown here (see supplement Fig. S3).

Figure 9: Daily Aitken- and Accumulation-mode PNC (in cm $^{-3}$ ) from the November 20 simulation. Panels (a)–(c) show results from the NoRIN, RIND $_{\sigma}$ , and RIND $_{\sigma}$ –NoRIN cases, respectively. For visual consistency, an upper limit of 10,000 cm $^{-3}$  is applied to the color scale in the PNC plots. However, the Aitken-mode PNC in the RIND $_{\sigma}$  simulation (panel b) frequently exceeds this limit. Panel (c) therefore provides a clearer depiction of regions with elevated PNC concentrations.





## 5. Discussion and Conclusion

## 5.1 Mechanistic insights and model implications

This study provides laboratory and modeling evidence supporting Recondensation-Induced Nucleation (RIN) as a physically plausible and atmospherically relevant pathway for new particle formation in urban environment. Laboratory experiments demonstrated that nucleation-mode particles emerge only when ambient aerosols are present in the intake air, confirming that pre-existing aerosols can undergo in-cylinder vaporization and subsequent re-nucleation during exhaust cooling. This process effectively converts accumulation-mode mass into Aitken-mode number, consistent with the observed particle size distribution (PSD) shifts in both engine exhaust and regional simulations.

Parcel-model simulations further elucidated the underlying microphysics of RIN. The results reveal a self-limiting nucleation process, whereby rapid particle formation enhances condensation and coagulation sinks that suppress further number growth. The sensitivity of peak particle number concentration (PNC) to both aerosol loading and evaporation fraction exhibited a saturation behavior, explaining why exhaust PNC increases sub-linearly with intake aerosol abundance in the laboratory. These findings underscore that RIN represents a secondary transformation process constrained by thermodynamic and microphysical feedbacks, rather than a simple linear source of ultrafine particles.

# 5.2 Urban-scale effects and model evaluation

When implemented in the CMAQ regional air-quality model, the RIN mechanism substantially improved simulation performance for Aitken-mode PNC while maintaining PM<sub>2.5</sub> mass consistency. Without RIN, the model exhibited a 75% underestimation of PNC and a 21% overestimation of PM<sub>2.5</sub>. The opposite biases in PNC and PM<sub>2.5</sub> is likely a common behavior of current air-quality models. Including RIN reduced the PNC bias down to 14% in the most physically consistent configuration (RIN<sub>Dσ</sub>), while the PM<sub>2.5</sub> bias remained largely unchanged. This contrasting response highlights the decoupled behavior of mass and number, supporting the hypothesis that traditional models omit a key mechanism influencing particle number concentrations in urban environments.

The diurnal analyses revealed that RIN-driven PNC peaks coincide with morning traffic activity, while PM<sub>2.5</sub> peaks occur later, reflecting the time lag between number production and mass accumulation. The earlier rise and faster decay of simulated PNC relative to PM<sub>2.5</sub> are consistent with both observational patterns (Chen 1999) and the coagulation-sink self-limitation seen in parcel modeling. This temporal behavior confirms that RIN is directly tied to vehicle emissions and acts on short timescales, making it a critical process for urban ultrafine particle dynamics.

Spatially, the inclusion of RIN markedly increased Aitken-mode PNC over major traffic corridors and metropolitan centers while slightly reducing accumulation-mode concentrations. The modeled Aitken-to-accumulation ratio (4–5) matched observations at the Xitun site, reinforcing the physical realism of the parameterization. Without the RIN mechanism, the ratio falls far below unity. The limited impact on accumulation-mode particles reflects compensating processes: while some mass is transferred to smaller sizes via re-nucleation, condensational growth and coagulation partially restore the accumulation







population, maintaining overall mass balance.

# 5.3 Implications for air quality and health assessment

The findings carry important implications for both air quality modeling and public health assessment. Conventional emission inventories and model frameworks emphasize mass-based metrics (PM<sub>2.5</sub>), yet this study demonstrates that such approaches can miss major number-based processes. The RIN mechanism provides a mechanistic explanation for the long-standing low-PNC bias seen in regional simulations. Incorporating RIN thus enhances the predictive capability of urban-scale PNC, a parameter more directly linked to ultrafine particle exposure and associated health effects.

Furthermore, because RIN depends on ambient aerosol composition and humidity, it may vary across different climatic and emission regimes. Regions with high sulfate and ammonium contents—such as East and Southeast Asia—are particularly conducive to RIN due to the high volatility of these compounds at combustion temperatures and their strong hygroscopic properties upon cooling. Consequently, including RIN in emission and chemical transport models could be critical for accurate exposure assessments in densely populated areas with intense traffic emissions.

## 5.4 Limitations and future directions

Although the experimental and modeling results collectively support RIN as a key urban aerosol process, several uncertainties remain. The current laboratory setup used a single-engine configuration under steady operating conditions; thus, future studies should extend testing to diverse engine types, fuels, and transient driving cycles. More comprehensive chemical and optical characterizations (e.g., SP2 for black carbon, PTR-ToF-MS for condensable organics) are also needed to refine the composition and volatility assumptions underlying the RIN parameterization.

In the modeling domain, the parameterization currently assumes uniform mixing of ambient aerosols and exhaust flows and instantaneous vaporization, which may not fully capture plume-scale and microscale interactions near emission sources. Further coupling of RIN with real-time traffic emission models, sub-grid plume dispersion, and aerosol microphysics modules could improve representation of near-road environments. Finally, long-term monitoring of size-resolved number concentrations in multiple cities will be essential to evaluate RIN's regional significance and its potential contribution to

## 5.5 Conclusions

ultrafine particle exposure risk.

This study establishes a comprehensive, multi-scale framework linking laboratory observations, parcel modeling, and regional simulations to demonstrate the atmospheric relevance of Recondensation-Induced Nucleation (RIN). The key findings are summarized as follows:

- Laboratory evidence confirms that RIN occurs only when ambient aerosols are present in engine intake air, producing a strong nucleation mode while depleting accumulation particles.
- Parcel-model simulations reveal that RIN-driven nucleation is self-limiting, with condensation and coagulation sinks

- governing the peak PNC response.
  - Regional CMAQ simulations incorporating RIN significantly improve PNC predictions in Taichung, reducing underestimation from 75% to 14%, while maintaining realistic PM<sub>2.5</sub> levels.
  - RIN offers a physically grounded explanation for the commonly observed low-PNC/high-PM<sub>2.5</sub> bias in air-quality models.
- The mechanism's sensitivity to ambient composition, humidity, and traffic activity highlights its importance for urbanscale particle dynamics and health-relevant air-quality assessments.
  - Overall, RIN provides a missing link between combustion microphysics and ambient particle number concentrations.
     Incorporating this process into air-quality modeling frameworks represents a critical step toward more accurate predictions of ultrafine particle exposure, advancing both scientific understanding and public health protection.

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
