# Peer review of "From cylinder to city: How the recondensation-induced nucleation shapes urban aerosol number"

_EGUsphere, 2025_

## Author Comment (AC1)

**Reply to Reviewer 1**

This study conducted laboratory research on the nucleation mechanisms during the cooling process of motor vehicle exhaust, constructed mechanistic models, and empirically applied them in a chemical transport model. Although the authors have undertaken substantial work, the logical connections among the three parts of the manuscript remain somewhat unclear. Significant revisions are required before reconsideration for publication.

**Reply:**

Thank you for the constructive comments, which have helped us to improve the manuscript. We agree that the logical connections among the three components of the manuscript can be made more explicit. Accordingly, we have reframed the study as a hierarchical, scale-bridging framework that progresses systematically from controlled laboratory observations to mechanistic insights, and ultimately to atmospheric-scale implications. To improve clarity and highlight this integration, we have revised the final paragraph of Section 1 as follows:

"The study is structured as a hierarchical, scale-bridging framework that progresses from controlled laboratory evidence to mechanistic understanding and, ultimately, to atmospheric-scale implications. The engine-exhaust experiments in Section 2 establish the empirical foundation by demonstrating the critical role of ambient aerosols in new particle formation under realistic combustion conditions. These findings motivate the idealized air-parcel simulations in Section 3, which translate the experimental conditions into a mechanistic framework that explicitly resolves particle evaporation during heating and nucleation during subsequent cooling. The parcel model explains the observed relationship between ambient aerosol loading and the resulting particle number concentration, and further examines the effects of incomplete particle evaporation. Insights gained from both the experiments and parcel simulations are then synthesized into a RIN parameterization implemented in the CMAQ regional air-quality model in Section 4. This final step assesses the atmospheric relevance of RIN by evaluating whether a mechanism identified at the laboratory scale can account for the long-standing underestimation of particle number concentrations in urban simulations without substantially altering $PM_{2.5}$ mass."

**Major comments:**

(1) In the introduction, the review of research progress on RIN is insufficient. It only briefly outlines various reasons that may lead to underestimation in particle size

spectrum simulations, without introducing the mechanistic research advancements specifically related to RIN, which is the focus of this study.

**Reply:**

Thank you for pointing this out. We agree that the original introduction did not sufficiently highlight prior mechanistic research related specifically to the RIN mechanism. In fact, progress on this topic has been relatively limited to date, which is a key motivation for the present study. Nevertheless, we have revised the introduction to more explicitly summarize the existing literature that has discussed RIN-related processes. Specifically, we now note that the RIN mechanism has been suggested to contribute to contrail formation from cryoplanes that do not emit combustion aerosols due to the use of liquid hydrogen (Ström and Gierens, 2002; Gierens et al., 2008; Lee et al., 2010), and that it has also been invoked to explain anomalous increases in particle number during dilution tunnel experiments (Lombaert et al., 2006). The relative scarcity of research beyond these specific cases highlights a significant gap in our mechanistic understanding of RIN and its broader air-quality impacts—a gap that the integrated experimental and modeling approach of this study is specifically designed to fill.

(2) In section2, Is the use of SMPS alone sufficient to characterize the number concentration of nucleation-mode particles? Nucleated particles often exist in large quantities below 3 nm, while the detection limit of SMPS starts at approximately 11 nm. This likely leads to an underestimation of particle counts at 11 nm and below.

**Reply**:

The reviewer raises a valid point regarding the detection limit. While the critical size of nucleation is indeed in the sub-3 nm range, several factors suggest that the SMPS (starting at 11 nm) captured the significant portion of the nucleation mode in this study. First, our measured PSDs consistently show a sharp decrease in number concentration as they approach the 11 nm lower limit. If a substantial population of smaller particles existed, we would expect to see an increasing trend toward the lower detection threshold. Second, the approximately 20-second residence time in the sampling tubing allows for significant particle growth via condensation and coagulation before the exhaust reaches the SMPS. This duration is sufficient for the majority of nucleated particles to grow beyond 11 nm. We have added a discussion of these sampling considerations and their impact on the captured PSDs to the end of Section 2.2.

(3) In section3, the experiments in section2 were conducted using gasoline engines to quantify the nucleation process during exhaust cooling. Is this mechanism

applicable to diesel vehicles? At the very least, this should be discussed. In real-world emissions, non-road mobile machinery, diesel vehicles, and even ships often use fuels with higher sulfur content. If the sulfuric acid-water binary nucleation mechanism can explain particle nucleation during cooling, these sources with higher fuel sulfur content might be more representative than gasoline vehicles. And what about the volatile organic.

**Reply:**

In principle, the RIN mechanism is applicable to all combustion processes that ingest ambient air containing pre-existing aerosols. However, the relative importance of RIN depends on the presence and strength of other nucleation pathways associated with fuel-derived combustion products. For on-road diesel vehicles, fuel sulfur content is generally comparable to that of gasoline in most developed regions, as both fuel types are regulated under ultra-low sulfur fuel standards (e.g., ≤10 ppm by mass in the European Union and Taiwan). Under such conditions, fuel-sulfur-driven nucleation in diesel engines is expected to be similar in magnitude to that in gasoline engines, suggesting that the RIN mechanism should operate in a broadly comparable manner for both vehicle types. In contrast, fuels used in marine shipping and some aviation applications may contain sulfur at levels of several thousand ppm, which can lead to strong sulfuric-acid-driven nucleation during exhaust cooling. In such cases, fuel-derived sulfate production may be comparable to or exceed that associated with RIN, and both sources would need to be considered jointly when estimating new particle formation. These combustion sources, however, are not the focus of the present study, which is centered on urban environments dominated by on-road traffic. We have added relevant discussion of these issues in Section 5.4 of the revised manuscript.

(4) Does the current mechanistic model lack consideration of the role of volatile organic compounds and semi-volatile organic compounds in nucleation during the condensation process? What impact would this have on the established mechanistic module and application of air quality model?

**Reply:**

Nucleation involving volatile organic compounds (VOCs) is another possible pathway. Although sulfuric acid has long been regarded as the primary driver of nucleation, recent studies show that highly oxygenated organic molecules can act as important contributors or even dominant agents in some environments. Nevertheless, our purified-air experiments indicate that particle production from fuel-only combustion—including VOC emissions from gasoline combustion—is much weaker than that associated with the RIN mechanism under our experimental conditions. We

note, however, that this conclusion is based on gasoline-engine experiments only. The potential significance of VOC-induced nucleation in diesel engines or other combustion sources remains uncertain and warrants further investigation.
We have added relevant discussion of these issues in Sections 5.4 of the revised manuscript.

(5) The validation of the numerical simulation is limited. Firstly, the validation of meteorological simulation is missing, making it difficult to confirm whether the underestimation of simulated concentrations is due to biases in the meteorological simulation.
**Reply:**
We appreciate this comment. To address the concern regarding meteorological validation, we have added new material to the Supplementary Information (Figs. S2 and S3), which compare simulated and observed synoptic-scale weather patterns as well as time series of key local meteorological variables. These comparisons indicate that the meteorological simulations exhibit some biases; however, their magnitudes are comparable to those typically reported in routine weather forecasts by our meteorological agency.
We note that one of the most influential meteorological factors for air-quality simulations in the study region is the planetary boundary layer height (PBLH), which strongly controls pollutant dilution and near-surface concentrations. Unfortunately, direct observational data for PBLH were not available during the study period, preventing a quantitative evaluation of this variable.

(6) Validation for gaseous precursors of PM2.5 such as SO2, NO2 and O3 is absent. Additionally, validation for related components is lacking, making it challenging to quantify the bias in simulated PM2.5 mass. This should be supplemented.
**Reply:**
Thank you for the suggestion to include validation of gaseous precursors and $PM_{2.5}$. In response, we have added comparisons between simulated and observed $SO_2$, $NO_2$, $O_3$, and $PM_{2.5}$ concentrations in the revised Supplementary Information (Fig. S4), along with brief discussions of the associated biases as follows:
Overall, simulated $SO_2$ and $NO_2$ concentrations are lower than those observed at the Xitun station. In addition to uncertainties in the meteorological fields, a likely contributing factor is the treatment of emissions in CMAQ, which spatially distributes point and line sources over the 2 × 2 km grid, thereby smoothing sharp near-road concentration gradients. This effect is particularly pronounced at Xitun, which is located in a dense urban setting adjacent to major roads and highway interchanges.

In contrast, biases in simulated $SO_2$ and $NO_2$ are smaller at the Fengyuan station, which lies in a transitional zone between the urban core of Taichung and the more rural northeastern region and is therefore less directly influenced by intense traffic emissions.

Simulated $O_3$ concentrations show better overall agreement with observations, but exhibit notable nighttime overestimation, especially at Xitun. This bias is likely related to insufficient NO titration when NO emissions are instantaneously diluted over a model grid cell. Unfortunately, routine observations of $PM_{2.5}$ chemical composition were not available during the study period, preventing a more detailed evaluation of individual $PM_{2.5}$ components.

(7) It is recommended to supplement the time-series simulation of particle number concentration and particle size distribution. Presenting only statistical average results lacks persuasiveness.

**Reply:** Thank you for the suggestion. To strengthen the evaluation, we have added time-series simulations of particle number concentration and particle size distribution for the whole simulation period, which provide a more detailed comparison beyond the statistical averages presented in the main text. Because the time series exhibit substantial day-to-day and diurnal variability, the resulting patterns are highly fluctuating and not easily summarized concisely in the main text. We therefore present these results in the Supplementary Information (Figs. S5 and S6).

**Specific editorial issues:**

- Line 276: "can be" is repeated.
**Reply:**
The duplicate phrase has been removed.

- Line 310: The abbreviation PNC and its full form appears redundantly; similar issues occur with PSD and RIN.
**Reply:**
We thank the reviewer and have standardized the use of acronyms throughout the manuscript to eliminate redundancy.

---

## Author Comment (AC3)

**Reply to Reviewer 2:**

The authors propose a new mechanism affecting particle number concentration and conduct extensive work from laboratory experiments to modeling, which may help improve the simulation of particle number concentrations. However, several statements and assumptions lack sufficient support. Major revisions are required before the manuscript can be reconsidered for publication.

**Reply:**

We thank the reviewer for their constructive assessment. We acknowledge that the support for certain assumptions needed to be more explicit. Accordingly, we have extensively revised the manuscript to provide the necessary evidence and theoretical foundations. Each of the reviewer's specific concerns is addressed in detail in the following sections.

**Comments**

(1) Lines 47–50: the authors state that "In urban areas, the underestimation of PNC is unlikely to be caused by weak/missing nucleation pathways, because the high concentration of existing aerosols will efficiently deplete precursors (e.g., sulfuric acid), leaving the nucleation process inconsequential." This statement is insufficiently supported. Existing studies have shown that nucleation processes, particularly previously underappreciated acid–base nucleation, are a major source of particle number concentration, and that sulfuric acid concentrations in urban environments can remain high and are still highly sensitive in controlling nucleation rates and particle number concentrations.

**Reply:**

Thank you for this important comment. Our original statement was partly based on both observational and modeling evidence specific to the study region, where PSD measurements seldom exhibit the characteristic "banana-shaped" evolution associated with regional new particle formation (NPF) events. For example, measurements at the Xitun station show only one such event during November 2020, and our CMAQ simulations likewise indicate negligible contributions from atmospheric NPF during the study period (see Fig. S6). In addition, several previous studies have reported that, in heavily polluted environments, nucleation contributes relatively little to particle number concentrations because high pre-existing aerosol loadings strongly deplete precursor vapors through condensation and coagulation. Taken together, these findings suggest that under the heavily polluted conditions frequently encountered in the study area, classical atmospheric nucleation pathways

play a limited role in controlling PNC.

We agree, however, that this statement should not be generalized to all urban environments, particularly given growing evidence that acid–base nucleation and high sulfuric acid concentrations can drive significant NPF under certain urban conditions. To avoid overgeneralization, we have revised the text to adopt a more nuanced formulation that is explicitly tied to environments with substantial pre-existing aerosol loadings. The revised text now reads:

"New particle formation (NPF) tends to be weak in environments—such as the study area—where substantial pre-existing aerosol concentrations act as strong condensation sinks for precursor vapors (Dal Maso et al., 2002; Manninen et al., 2010; Jeong et al., 2010; Qi et al., 2015; Salma et al., 2016), and can even be largely inhibited under heavily polluted conditions (Guo et al., 2014; Gani et al., 2020). These findings suggest that, under the heavily polluted conditions frequently encountered in the study area, classical atmospheric nucleation pathways play a limited role in controlling particle number concentrations. Consequently, neither the measurements nor the air-quality simulations presented below reveal significant NPF events."

We believe this revision more accurately reflects both the literature and the scope of our observations, while acknowledging that nucleation processes may remain important in other urban settings (as discussed in Sections 3.1 and 5.4).

(2) Line 73: the authors should provide supporting data, such as the proportion of motorcycles in the total vehicle fleet or their contribution to emissions, to justify the representativeness of motorcycles in this study.

**Reply:**

Thank you for the suggestion. To justify the representativeness of motorcycles in this study, we have added supporting information to the manuscript. Specifically, motorcycles account for approximately 60–65% of registered vehicles in Taiwan, and their proportion is similarly high in Taichung City. In addition, motorcycles contribute disproportionately to urban emissions due to their large numbers, frequent stop-and-go operation, and high activity in near-road environments. According to the latest Taiwan Emission Data System (TEDS 11.0/12.0), motorcycles are a primary contributor to urban air pollution, accounting for nearly 40% of total $PM_{2.5}$ emissions in the studied area of Taichung City. These statistics support the use of motorcycles as a representative platform for investigating traffic-related particle formation processes in this region.

(3) Line 77: extrapolating experimental results from a single motorcycle to all

motorcycles and passenger vehicles may introduce significant uncertainty. An uncertainty analysis is needed to discuss how differences in vehicle type, engine technology, and operating conditions may affect the results, and the applicability range of the proposed parameterization should be clearly stated.

**Reply:**

We appreciate this comment and agree that extrapolating results from a single engine configuration introduces uncertainty. The primary physical requirement for the RIN mechanism is that in-cylinder or exhaust temperatures are sufficiently high to vaporize pre-existing ambient aerosols, thereby supplying condensable vapors that can re-nucleate during exhaust cooling. In this respect, both motorcycles and passenger vehicles operate at peak combustion temperatures that are well above the thresholds needed for aerosol vaporization, and in fact exceed the temperatures used to demonstrate RIN in earlier furnace experiments (Chen, 1999) and in our parcel-model simulations. This provides confidence that the fundamental RIN mechanism should be active across a wide range of gasoline-powered on-road vehicles.

We acknowledge, however, that vehicle type, engine technology, and operating conditions (e.g., engine load, exhaust after-treatment, and cooling rates) can influence the quantitative strength of RIN by modifying factors such as evaporation efficiency, exhaust cooling timescales, and interactions with combustion residues. These factors may affect the magnitude of particle number production but are not expected to suppress the underlying mechanism itself. Accordingly, we interpret the parameterization developed in this study as representing the order-of-magnitude impact and qualitative behavior of RIN under typical urban driving conditions, rather than a vehicle-specific emission factor.

To address this concern, we have clarified in the revised manuscript (mainly in Section 5.4) the applicability range and limitations of the proposed parameterization, explicitly noting that it is most appropriate for gasoline-powered on-road vehicles operating under conditions common in urban environments. A more comprehensive quantitative assessment of how RIN varies across different vehicle classes, engine technologies, fuels, and driving conditions would require targeted experiments and is beyond the scope of the present study; we identify this as an important direction for future work.

(4) Line 162: the authors state that "new particle formation is dominated by $H_2SO_4$–$H_2O$ nucleation," which is inconsistent with the widely accepted understanding that acid–base nucleation is the dominant nucleation mechanism in urban boundary-layer environments. Moreover, the two references cited in support of this statement do

not appear to provide direct evidence for such a conclusion. The authors should provide clearer evidence to support this claim and explicitly discuss the uncertainties introduced by adopting only the $H_2SO_4$–$H_2O$ nucleation framework while neglecting acid–base nucleation, as well as how the presence of ammonia or semi-volatile organic compounds in urban environments may modulate the efficiency of the proposed RIN mechanism.

**Reply:**

We thank the reviewer for pointing out this important distinction. We agree that our original wording was imprecise. Because the RIN mechanism depends on the rapid cooling and recondensation of volatile precursors within the immediate exhaust plume, the nucleation conditions in the ambient urban boundary layer are not directly comparable to those found in combustion exhausts. Consequently, we have removed the misleading references and revised the first paragraph of this section to focus exclusively on the near-field plume thermodynamics relevant to RIN.

In the study region, sulfate and organic carbon are the dominant components of $PM_{2.5}$, with nitrate and ammonium also contributing substantially (Chou et al., 2022), indicating that acid–base (e.g., $H_2SO_4$–$NH_3$–$H_2O$) nucleation is certainly plausible in the ambient atmosphere. Nevertheless, our idealized parcel simulations demonstrate that binary $H_2SO_4$–$H_2O$ nucleation alone is sufficient to produce particle number concentrations comparable to those observed in the laboratory experiments. While the inclusion of base species such as ammonia or semi-volatile organic compounds would likely enhance nucleation efficiency, we do not expect this to fundamentally alter the qualitative behavior of the system, because new particle production under high vapor loadings rapidly becomes self-limited by condensation and coagulation sinks—a feature captured by our simulations and documented in previous studies. Importantly, the purpose of the parcel simulations is not to reproduce atmospheric nucleation pathways in urban air, but rather to provide a mechanistic interpretation of key laboratory observations, particularly the dependence of particle production on intake aerosol loading and the effects of incomplete particle vaporization during combustion. The parcel-model results are therefore used for qualitative interpretation only and are not directly applied in the CMAQ simulations.

To clarify these points, we have revised the opening paragraph of Section 3 and added explicit discussion of nucleation pathway uncertainties and combustion-related factors in Section 5.4 (paragraphs 2–3) of the revised manuscript.

(5) Lines 334–335: Although the simulations cover the period from 5 to 25 November 2020, the validation results shown in Figures 8 and 9 are limited to only two specific days (20 and 25 November). This limited validation makes it difficult to demonstrate

the robustness of the modeling framework over a broader time period. It is recommended to provide validation results over longer time scales and to supplement the evaluation with quantitative statistical metrics such as RMSE.

**Reply:**

We appreciate this comment and agree that validation over a longer time period strengthens the assessment of model robustness. In the main text, we focused on two representative days (20 and 25 November) to illustrate the underlying processes without obscuring the discussion of the RIN mechanism with detailed model–observation discrepancies, whose sources can be complex and difficult to disentangle.

To address the reviewer's concern, we have expanded the evaluation in the Supplementary Information by adding time-series comparisons of simulated and observed particle number concentration and particle size distributions for the broader simulation period from 5 to 25 November (Figs. S5 and S6). The root-mean-square error (RMSE) of PNC is now reported in the figure captions. These additional results demonstrate that the performance improvements associated with the RIN parameterization persist over the full simulation period, thereby supporting the robustness of the modeling framework beyond the two illustrative case days shown in the main text.

(6) The title of the manuscript should be further constrained to explicitly indicate that the proposed recondensation-induced nucleation is related to traffic or vehicle exhaust emissions. The current wording is overly generalized. Although "cylinder" may implicitly refer to engine combustion processes, this terminology is too implicit and not sufficiently clear for an atmospheric science audience.

**Reply:**

Thank you for this helpful suggestion. We agree that the title should more explicitly indicate that the proposed recondensation-induced nucleation mechanism is associated with traffic-related vehicle exhaust emissions. To address this concern and avoid potential ambiguity, we have revised the title to explicitly include "vehicle exhaust" as the following: "From cylinder to city: How recondensation-induced nucleation in vehicle exhaust shapes urban aerosol number." We believe this wording more clearly conveys the scope of the study to an atmospheric science audience while retaining the original conceptual framing.